# Genomic Insights on the Carbon-Negative Workhorse: Systematical Comparative Genomic Analysis on 56 *Synechococcus* Strains

**DOI:** 10.3390/bioengineering10111329

**Published:** 2023-11-18

**Authors:** Meiwen Qian, Xiao Han, Jiongqin Liu, Ping Xu, Fei Tao

**Affiliations:** School of Life Sciences and Biotechnology, Shanghai Jiao Tong University, Shanghai 200240, China; 119080930114@sjtu.edu.cn (M.Q.); smile56@sjtu.edu.cn (X.H.); 697_jiongqinliu@sjtu.edu.cn (J.L.); pingxu@sjtu.edu.cn (P.X.)

**Keywords:** *Synechococcus*, comparative genomic analysis, whole genome, temperature, salinity

## Abstract

*Synechococcus*, a type of ancient photosynthetic cyanobacteria, is crucial in modern carbon-negative synthetic biology due to its potential for producing bioenergy and high-value products. With its high biomass, fast growth rate, and established genetic manipulation tools, *Synechococcus* has become a research focus in recent years. Abundant germplasm resources have been accumulated from various habitats, including temperature and salinity conditions relevant to industrialization. In this study, a comprehensive analysis of complete genomes of the 56 *Synechococcus* strains currently available in public databases was performed, clarifying genetic relationships, the adaptability of *Synechococcus* to the environment, and its reflection at the genomic level. This was carried out via pan-genome analysis and a detailed comparison of the functional gene groups. The results revealed an open-genome pattern, with 275 core genes and variable genome sizes within these strains. The KEGG annotation and orthology composition comparisons unveiled that the cold and thermophile strains have 32 and 84 unique KO functional units in their shared core gene functional units, respectively. Each KO functional unit reflects unique gene families and pathways. In terms of salt tolerance and comparative genomics, there are 65 unique KO functional units in freshwater-adapted strains and 154 in strictly marine strains. By delving into these aspects, our understanding of the metabolic potential of *Synechococcus* was deepened, promoting the development and industrial application of cyanobacterial biotechnology.

## 1. Introduction

The oxygen-producing photosynthesis of cyanobacteria is one of the most profound physiological roles of microorganisms within the Earth’s environment and evolution of life. *Synechococcus* and *Prochlorococcus* are the two most dominant groups of cyanobacteria [1,2]. Compared to the latter, *Synechococcus* is a more versatile responder and widely distributed species, with diverse metabolic forms [3], playing a pivotal role in resource utilization and environmental protection, such as wastewater treatment, high-value product extraction, and biomass transformation [4], thus contributing to a dual harvest of economic and ecological benefits.

*Synechococcus* is a genus of cyanobacteria defined by its morphological characteristics [5]. Its size ranges from 0.2 to 2 μm. In 1979, Paul W. Johnson officially recognized it as “small unicellular cyanobacteria with ovoid-to-cylindrical cells that reproduce through binary traverse fission in a single plane and lack sheaths” [6]. *Synechococcus* exhibits extensive diversity in ecological habitats and can be found in various ecosystems, including some of the most extreme environments such as hot springs, the equator, and polar regions [7]. Many *Synechococcus* species have key advantages such as high biomass, fast growth rate, genetic editing capability, and potential for conversion into biofuels [8], making them ideal model organisms for promoting the development of new synthetic biology chassis. *Synechococcus* strain UTEX 2973 is a fast-growing cyanobacteria strain with good resistance to high temperature and huge potentials for the production of carbohydrate feedstocks, accumulating glycogen contents as high as 50% of dry cell weight independent of nitrogen depletion [9]. It represents a promising candidate for use as a synthetic biology chassis. Previously, *Synechococcus* was tested and optimized as chassis for PHB production [10]. Furthermore, *Synechococcus elongatus* PCC 7942 was genetically modified to include the heterologous pathway for PLA production, making it a suitable chassis for bioplastic synthesis [2]. Of note, synthetic biology provides powerful technical support for the industrialization of *Synechococcus*. Emerging synthetic biology tools, such as regularly interspaced, clustered, short palindromic repeats (CRISPR)/cpf1, riboswitches, and metabolic network reprogramming circuits, have accelerated the industrial applications of these tools [11]. By improving the host characteristics, researching and implementing pathway engineering strategies, and enhancing target products, synthetic biology technology is expected to push the industrialization of *Synechococcus* to a new level [12].

Since the sequencing of the first strain of *Synechococcus* sp. WH 8102 in 2003 [13], complete genomic sequences and annotation information of this genus have been emerging. In 2009, scientists successfully assembled the first complete *Synechococcus* strain CC9902, with a genomic size of 2.24 Mb. In 2020, the sequencing of the strain CBW1006 was completed; this strain had the largest genome at 3.86 Mb among the sequenced strains. *Synechococcus* comprises relatively small genome sizes and complexities [14], consisting of a single circular chromosome without plasmids, with genes necessary for survival and photosynthesis, allowing for adaptive advantages for survival [15].

However, due to the complexity and instability of their growing environments, industrial production faces many challenges. Synthetic biology attempts to achieve design goals by reconstructing metabolic networks, but the complexity of metabolic networks increases the uncertainty of this process. Comparative genomic analysis can identify key nodes and regulatory factors in metabolic networks, guiding the rational reconstruction of metabolic networks and providing theoretical support for improving the environmental adaptability of *Synechococcus*.

Based on comprehensive genomic sequence analysis, 56 *Synechococcus* strains were compared in this study on the premise of clarifying genetic relationships, the adaptability of *Synechococcus* to the environment, its reflection at the genomic level in terms of the pan-genome, and a comparison of functional genes was explored. Moreover, six representative strains with different temperature and salinity preferences were selected, and some relevant key metabolic pathways and genes were identified, laying the foundation for research on salt and temperature adaptation mechanisms.

## 2. Materials and Methods

### 2.1. Synechococcus Strains and Database Accession Numbers

A total of 56 *Synechococcus* strains with complete assembly were accessed using the accession numbers listed in Table 1 for their genome characteristics.

### 2.2. Construction of Phylogenetic Trees

The phylogenetic analysis was conducted using the complete genome sequences of 56 *Synechococcus* strains as operational taxonomic units and established on GTDB (https://gtdb.ecogenomic.org/ accessed on 18 August 2023).

The relative evolutionary tree was constructed using MAFFT V7 (https://mafft.cbrc.jp/alignment/software/ accessed on 18 August 2023) with 16S rRNA, peroxiredoxin, and cytochrome C oxidase subunit I as evolutionary markers. During the alignment process, the “automated1” parameter option was selected, and the software automatically calculated and selected the optimal evolutionary distance model to obtain a tree file. The tree file format was visualized using the IQ-TREE web server (http://iqtree.cibiv.univie.ac.at/ accessed on 18 August 2023), and modified using the online tree aesthetic tool iTOL (https://itol.emnl.de/ accessed on 23 August 2023).

### 2.3. Pan-Core Genomes Analysis

The pan-genomic analysis was performed using the BPGA (Bacterial Pan Genome Analysis tool) software on selected strain genomes. Additionally, gnulpot was installed using USEARCH to perform power law regression analysis and generate a pan-core gene trend plot. The functional annotation module of BPGA was then utilized to visualize the COG functional annotation results by inputting genome sequences and COG data into GenBank format files, with specific parameters set such as an E-value threshold of 10^−5^ and the USEARCH clustering algorithm with an identity value of 0.5.

### 2.4. Analysis of Genes Exclusively Related to Temperature and Salinity

Six strains have been selected for comparative analysis that are representative, highly studied, and well-characterized in terms of temperature and salinity. These strains exhibit exceptional performance and robust tolerance, providing strong support for further research and applications. After enrichment and KEGG pathway analyses, the sequence information of the core genome was extracted and annotated in the KEGG database. The KO identifiers for common gene functional units were obtained and mapped to the KEGG pathway database to find metabolic pathways and the key genes related to environmental adaptation. These KO identifiers for temperature and salinity preference strains were organized into a text document format and uploaded to the online tool Venny (https://bioinfogp.cnb.csic.es/tools/venny/ accessed on 24 August 2023). This tool displays the intersection and difference set of multiple sets of data in a visual and concise manner, with the difference set representing unique gene functional units of temperature- and salinity-adaptive types of strains, predicting the possible association with environmental adaptation.

## 3. Results

### 3.1. General Features of 56 Synechococcus Strains

The genome size of these strains ranges from 2.11 to 3.86 Mb, as shown in Table 1. They have various G + C contents, ranging from 40.6% in PCC 7502 to 68% in RSCF101, suggesting a higher diversity of genomes. The number of marine strain genomes sequenced is much greater than that of the freshwater strains. Compared with marine isolates (≈2.63 Mb, 57.57%), freshwater isolates (≈3.1 Mb, 50.25%) have relatively larger genomes, but a lower GC content. The size of protein coding sequences (CDS) ranges from 2288 (*Synechococcus* sp. WH8109) to 3654 (*Synechococcus* sp. BMK-MC-1), and there is a positive linear relationship between CDS and genome sizes (Figure 1). Compared to other genera in cyanobacteria, the genome size of *Synechococcus* is relatively small but the gene density is relatively high.

### 3.2. Phylogenetic and Comparative Genome Analyses

The information of 56 *Synechococcus* strains, including their name, geographical origin, and collection depth, is shown in Table 2. These strains can be divided into subclusters, with different species having distinct geographical distribution patterns (Appendix A).

Upon constructing evolutionary trees using different molecular markers, the findings showed that there is a certain geographical environmental correlation between different strains, with those sharing the same habitat tending to cluster together (Figure 2). All confidence levels were very high, and when comparing mutual verification, it was observed that the genetic relationships were consistent, indicating the reliability of using conservative markers for evolutionary analysis. Among them, the evolutionary tree based on peroxidase construction highlighted the decisive role of intrinsic features in the protein sequence [16]. These trees depicted different aspects of the influence of the environment and genetic factors on the evolution process of the strains and specific enzymes, revealing the complex evolutionary history of *Synechococcus*.

### 3.3. Core and Pan Genomic Analyses of the 56 Synechococcus Strains

In pan-genomic studies, the power law model [19] can be used to determine whether a strain’s genomic data belong to a conservative pan-genome or an open pan-genome. The power law regression analysis of 56 *Synechococcus* strains performed in Figure 3 showed that the functional adaptability value of the 56 strains was 0.69 (<1), indicating that the pan-genome is still open. This indicates that the genomes of *Synechococcus* strains are highly plastic and may acquire new genes more easily, making them more adaptable to various complex environments and leading to a wider distribution. Their core genomes, on the other hand, are relatively conservative: as the number of analyzed genomes increases, more gene acquisition and loss events occur among different strains, resulting in a gradually decreasing number of core genes. When the number of analyzed strains increases to 56, the total number of gene families reaches around 30,000, and the final number of core gene families stabilizes at around 275 (Appendix A). The COG functional annotation of the specific genomes displayed in Figure 4 showed that the most common functions in the core genome of *Synechococcus* are related to the metabolism (52.8%), and information storage and processing (34.1%), while the distribution of accessory and unique gene functions was similar, with metabolic and information-processing functions accounting for 32.46% and 26.49%, respectively. Intracellular biological processes and signaling mechanisms accounted for 15.81% and 19.14%, respectively. When poorly characterized, these two accounted for 27.61% and 27.41%, respectively.

### 3.4. Temperature Adaptation Mechanism of Synechococcus and Comparative Genomics

#### 3.4.1. Growth Temperature and Structural Characteristics of the *Synechococcus* Genome

Based on their minimum tolerance for growth temperature, these strains could be further classified into cold-adapted, warm-adapted, and thermophilic strains (Table 3).

The growth temperature range and optimal growth temperature of warm-adapted strains are moderate; they cannot survive at temperatures below 10 °C and the optimal growth temperature is generally around 30–40 °C. The lowest growth temperature of cold-adapted strains is below 16 °C. Compared to warm-adapted strains (3.4 Mb, GC 49%), the genome sizes of 3.0 Mb for thermophilic strains and 2.6 Mb for cold-adapted strains are relatively small, but the GC contents (about 60% for thermophilic strains and about 58% for cold-adapted strains) are higher. Almost all cold-adapted strains can survive or culture at around 20 °C, indicating that all *Synechococcus* strains should have certain heat resistance.

#### 3.4.2. Temperature Adaptation Mechanisms

The total number of core KO identifiers is 2305 for thermophilic strains, 2704 for warm-adapted strains, and 1383 for cold-adapted strains, as displayed in Figure 5. After further comparison of the core KO identifiers of different temperature-type strains, we concluded that the gene functional units found only in cold-adapted and thermophilic bacterial strains contain 32 and 84, respectively.

#### 3.4.3. Thermal Adaptation Strategies

The thermophilic strains possess 84 specific core gene functional units (Appendix A), which involve various metabolic and disease pathways, such as cancer and bacterial resistance. The core metabolic functions of cold-adapted strains contain 32 unique gene functional units (Appendix A), with the most abundant being the core metabolic pathway (Figure 6). Each KO functional unit contains one or more key genes related to thermal adaptation, and their unique key genes can be regulated through the following strategies: (1) Heat shock protein expression. In response to high temperature environments, the primary adaptive response of all organisms is a heat shock response [20]. An increased expression of *SEC63* in response to high temperatures can help correct misfolding errors caused by high temperatures [21]. (2) Molecular repair mechanisms. CDC48 utilizes ATPase activity to facilitate the assembly and disassembly of protein complexes, thereby clearing misfolded proteins in various organelles such as the nucleus, cytoplasm, endoplasmic reticulum, and mitochondria [22]. An MPG protein initiates base excision repair by cutting off the glycosyl bonds of numerous damaged bases, thus repairing DNA damage caused by high temperatures [23]. (3) Accumulation of solutes for protection. Under high-temperature conditions, bacteria accumulate solutes to balance the water loss caused by high temperatures and help cells maintain osmotic balance, thus increasing their survival rate [24]. TreY, TreZ, and MTTase possess thermotolerance properties and can maintain stable structures and catalytic activities at high temperatures, helping bacteria adapt to glucose metabolism and energy utilization within high temperature environments. The cysA-encoded sulfate transport system provides sulfur sources for the synthesis of hydrogen sulfide and helps thermophilic bacteria aggregate compatible sulfate solutes, improving the stability of cells in high-temperature environments [25]. The specific OpuA, OpuB, OpuC, and OpuD transport systems in thermophilic bacteria have been extensively studied for their stress protection functions. The uptake of OpuA transporter mediates the high-affinity uptake of glycine betaine and proline betaine [26]. The substrate-binding protein of OpuC [27] can recognize a wide range of compatible solutes such as choline, botulinum toxin, and glycine betaine.

#### 3.4.4. Cold Adaptation Strategies

Typically, organisms adapt to low-temperature environments by regulating the expression of cold shock proteins. However, no common cold-shock-protein-encoding genes were detected in the genome of the cold-adapted *Synechococcus* strains studied, such as *cspA*, *cspB*, *cspC*, or *cspG*. This indicates that *Synechococcus* does not rely on cold shock proteins for survival under low temperatures and employs alternative pathways and molecular mechanisms to achieve low temperature adaptation. For instance, it involves fatty acid synthesis and unsaturation modulation related to membrane fluidity, solute accumulation, and the stabilization of protein structures.

The increase in unsaturated fatty acid concentration enhances membrane fluidity at low temperatures. LcyB can convert lycopene to carotene, enhancing the tolerance of strain due to salt, drought, and oxidative stress [28]. LcyB also catalyzes the cyclization of lycopene to produce desaturated carotenoids such as aromelin and naranjine, which can be converted into precursors of unsaturated fatty acids in cell membranes [29].

The rational regulation of glycogen synthesis and breakdown is one of the crucial mechanisms for spirulina to adapt to low-temperature conditions. GlgX and GlgP proteins degrade glycogen; the absence of glgX leads to excessive glycogen accumulation [30]. Under low-temperature conditions, glycogen acts as a storage carbon source and energy source. The upregulation of glycogen synthesis pathways allows excess glucose to be stored temporarily to prevent excessive solutes from inhibiting metabolic enzyme activity. Increasing glycogen accumulation also helps balance water entry due to low temperature, while inhibiting glycogen-decomposition-related enzyme activities to reduce glycogen consumption [31].

Stabilizing protein structure is among the strategies employed by cold-adapted *Synechococcus* strains to respond to low-temperature stresses. Pex5 plays a critical role in targeting proteins to the peroxisome for degradation. Even under low-temperature conditions, Pex5 retains its transport and import functions to help correctly fold and localize enzymes within the peroxisome [32]. RhlE1, an example of this family, can still participate in protein folding under suboptimal growth temperatures [33]. *DNAJC3* is a molecular chaperone localized to the endoplasmic reticulum that transiently binds to newly synthesized proteins, especially the regions with unstable structures, helping maintain proper protein synthesis and prevent the incorrect folding of susceptible domains [34].

### 3.5. Salt Tolerance and Comparative Genomics in Synechococcus

#### 3.5.1. Genome Features of *Synechococcus* with Different Salinity Growth Conditions

Among the 56 strains, salt tolerance ranges, optimal salt levels, and salt tolerance categories were studied in 27 *Synechococcus* strain genomes, as listed in Table 4. The genome size of freshwater strains was between 2.67 and 3.72 Mb (average of 3.06 Mb); for euryhaline strains, it was between 2.44 and 3.86 Mb (average of 3.28 Mb); and that of strictly marine strains was between 2.22 and 3.35 Mb (average of 2.58 Mb). The GC content of freshwater strains was between 40.6% and 55.5%, while that of salt-tolerant strains was between 62.6% and 67.1%. These values indicate that of euryhaline strains tend to have larger genome sizes, beneficial for encoding more salt resistance-related proteins. Strictly saline strains have relatively higher GC contents, which helps maintain the stability of their genomes.

#### 3.5.2. Salinity Adaptation Mechanisms

Compared with the functional units of the core genes of salt-loving bacteria, 65 KO functional units were specific to freshwater stains, and 154 KO functional units were specific to strictly marine strains (Figure 7). More KO units were identified in the core and specific units, which may be related to the fact that strictly marine stains require more genes to support their adaptation to high-salinity environments.

According to the list of unique 154 KEGG pathways of strict marine strains (Appendix A), the highest number of pathways belongs to the ABC transporter pathway of metabolism pathway. The relatively higher number of pathways is related to the synthesis and accumulation of solutes such as glycogen and lipids. The freshwater strains have 65 unique keg core identifiers (Appendix A), which represent different pathways, including the majority of the core and sugar metabolic pathways. This demonstrates strong environmental adaptability by regulating energy metabolism, nutrient absorption mechanisms, and oxidative redox homeostasis (Figure 8); freshwater strains adapt to low-salt or salt-free environments.

#### 3.5.3. High-Salt Adaptation Strategies

Analyzing the key genes involved in specific functional units of obligate aerobes, strictly saline strains can adapt to high salinity via the following strategies:Synthesis of compatible solutes

The accumulation of compatible solutes is a common defense measure used by bacteria to resist harmful effects caused by high salinity and tolerate high-salinity environments. On one hand, this method can alleviate high salinity pressure, and on the other hand, these compatible solutes can be quickly synthesized and degraded [35]. The *CodA* gene can convert choline into betaine glycine, providing tolerance to salt stress [36]. Amt is an important enzyme in the betaine metabolism pathway. It catalyzes the conversion of betaine from methionine to betaine glycine, which is a compatible solute that can resist salt stress [37]. Glycerol acts as a compatible solute and enhances tolerance to various non-biotic stresses. Glycerol metabolism is mediated by *glp* genes. Under conditions of glycerol phospholipid metabolism, the encoding *glpA* gene is highly upregulated, leading to an increased accumulation of glycerol in cells [38].

2.Antioxidant defense

Salinity induces changes in osmotic potential in cells by reducing the water potential. Subsequently, the accumulation of ions in cells disturbs ion homeostasis and leads to changes in membrane permeability, affecting essential ion absorption. These disturbances caused by salinity lead to a series of metabolic disorders followed by reactive oxygen species (ROS) generation [39]. Glutathione peroxidase (GPX) is an antioxidant enzyme that exhibits high expression under different types of environmental stressors such as bacterial infection, contact with heavy metals, or high concentrations of salt [40]. Under salt-stress conditions, genetically modified strains show better anti-salinity than non-genetically modified individuals, indicating that overexpression of the GPX gene can effectively protect the strain from harm caused by salinity and promptly eliminate excess hydrogen peroxide and lipid peroxides. Overexpression of the GPX gene does not affect normal growth of the strain, but enhances its anti-salinity ability [41].

3.Ion transport and distribution

Ion transport is a crucial step in controlling cellular uptake and subsequent storage, reduction or export that is essential for the growth of strictly marine strains. Phosphate is a basic nutrient required for all living organisms. All genes responsible for using phosphorus are located in the phosphorus transport system (phnC) [42]. Under high-salinity conditions, increasing Na^+^ intake leads to an elevated osmotic potential, which helps balance the osmotic energy required for cell growth via an addition phosphorus [43]. Under salt stress conditions, PstA can act as a sensor for monitoring changes in intracellular phosphorus concentration due to changes in its expression and activity [44]. The transport process of nitrate and nitrite mainly involves MFS (facilitated diffusion superfamily) transporters [45]. To avoid toxicity caused by nitrate, it undergoes assimilatory reduction and active transport processing [46].

#### 3.5.4. Freshwater Adaptation Strategies

In terms of energy metabolism, *Synechococcus* utilizes methane and other organic substances in freshwater to generate energy. For example, it participates in methane metabolism through the beta subunit of the hydrogenase enzyme coded by *frhB* [47], and the two subunits coded by *cofG* and *cofH* form a methane monooxygenase that participates in the pathway [48]. Furthermore, the key enzymes in the glycolysis pathway, such as 3-phosphoglycerate dehydrogenase (GAPDH) and fructose-1,6-bisphosphate aldolase (FBPase), contribute to energy production [49].

In terms of nutrient absorption, the alga utilizes phosphorus transport systems regulated by pstS [50], sulfate transport systems composed of CysP, CysU, and CysW [51], branched amino acid transport systems regulated by *livF*, *livG*, *livH*, and *livM* genes [52], and carbonic bicarbonate transport system including *cmpA*, *cmpB*, and *cmpC* genes [53] to uptake scarce nutrients. These systems help cells absorb essential amino acids, sulfates, and bicarbonates to meet their freshwater survival needs.

Regarding oxidative redox regulation, the alga protects cells from high salt or oxidative stress through enzymes with antioxidant functions encoded by the *katE* and *CAT* genes [54] and superoxide dismutase SOD2 [55]. These mechanisms help maintain a stable redox state within cells, resisting external environmental pressures.

## 4. Discussion

This study compared and analyzed the genome sequence of *Synechococcus*, elucidating the genomic features and relationship with environmental adaptability from multiple perspectives. The *Synechococcus* genome is highly diverse, with different strains forming specific genomic compositions, key genes, and metabolic pathways based on temperature and salinity environments. A systematic phylogenetic and pan-genome analysis was conducted on 56 strains at the whole-genome level. The results showed that these strains have an open-genome pattern, with 275 core genomes and variable genomes of varying sizes. The phylogenetic relationships of different strains are related to their survival environment.

By isolating only a single variable (either temperature or salinity), a complete picture of the fitness of *Synechococcus* populations with different genotypes relative to natural variations in both of these conditions might not have been obtained. Furthermore, environmental parameters other than temperature and salinity (e.g., nutrient availability, pH, etc.) may influence the overall fitness of these organisms and thus affect their distribution. The gene families and pathways found in this study may reveal further evidence of adaptation to temperature (and salinity) and could give insight into other environmental factors affecting the adaptations of *Synechococcus*.

The growth of *Synechococcus* strains is related to temperature, which is a major environmental factor controlling photosynthetic rate and biogeography [56]. The high genetic diversity in *Synechococcus* leads to their ability to tolerate a wide range of temperatures [57]. Temperature plays a crucial role in many industrial processes and chemical reactions. Thermophilic *Synechococcus* can reduce dependence on low-temperature environments in industrial fermentation processes, decrease industrial fermentation costs, and increase production efficiency. From the perspective of temperature adaptability, the genomic sequences of different types of strains were analyzed. The results showed that compared with warm-adapted strains, the genomes of cold-adapted and thermophilic strains underwent reduction. Through KEGG annotation and orthology (KO) composition comparisons, it was concluded that there are 32 and 154 unique KO functional units within the shared core gene functional units of the two types of strains, respectively, which rely on metabolic pathways such as the regulation of heat shock protein expression and membrane fatty acid composition for temperature adaptation. Extreme thermophilic or cold-adapted bacteria are highly interesting from an industrial processing perspective. The ability to grow at high or cold temperatures in bioreactors reduces the costs of cooling and prevents contamination by mesophilic spoilage bacteria [58]. *Synechococcus* can be used as a platform to introduce heat and cold adaptive genes identified in this study, enhancing its heat and cold tolerance. Additionally, utilizing *Synechococcus* as a chassis facilitates the production of cell factories after introducing these genes, offering innovative solutions for bioenergy, biologic medicine, environmental protection, and other fields.

An attractive strategy for controlling biological contamination is to increase the salinity of the growth media. Salt-tolerant *Synechococcus* is capable of adapting to seawater fermentation, and can be used to reduce production costs and simplify the process. Under different salinities, enriched populations are clearly different [59]. Most *Synechococcus* strains have adapted to long-term relatively stable salinity levels and cannot tolerate drastic changes in salinity [60]. Compared with freshwater host strains, most *Synechococcus* strains colonize marine environments and have greater resource utilization advantages. For example, PCC7002 and PCC11901 are not only rapidly growing, salt-tolerant, and heat-resistant [61,62], but also good chassis strains for industrial applications. From the perspective of salinity adaptability, among the freshwater-adapted strains, there are 65 unique KO functional units in the shared core genome that rely on energy metabolism, nutrient absorption, and osmotic pressure regulation for freshwater adaptation. For saline-tolerant strains, there are 154 unique KO functional units in the shared core genome that adapt to high-salinity environments through antioxidation, nutrient absorption, and transport regulation.

This study provides genomic-scale insights into the genetic basis of *Synechococcus*’ adaptation to different temperatures and salinities, offering a new perspective and abundant gene targets for understanding the environmental adaptability of cyanobacteria. Future research can expand upon several aspects by: (1) expanding the sample size by collecting more complete genome sequences of *Synechococcus*, hence enhancing our understanding of its diversity and evolutionary relationships; (2) deeply analyzing the predicted key genes and their metabolism regulation in temperature and salt stress responses, experimentally verifying their contribution to improving environmental adaptability, and aiding application development; (3) investigating the effects of different environments on *Synechococcus* phenotypes and genotypes, expanding our knowledge of microbial adaptability theory; and (4) attempting to modify relevant genes and optimize *Synechococcus*’ environmental adaptability. Genome size information helps clarify the metabolic characteristics of chassis cells and guide the design of synthetic biology based on their features.

The research prospects in the field of *Synechococcus* are vast. It not only deepens our understanding of the rules governing microbial environmental adaptability but also actively promotes their applications in environmental remediation, high-value product biosynthesis, and other areas, fully exploring and utilizing its potential value.

## 5. Conclusions

In this research, the complete genome sequences of 56 *Synechococcus* strains were selected as the analysis targets for conducting the comprehensive genomic investigation. Bioinformatics tools were employed to analyze and compare the genomic features of various strains from a comparative genomics perspective, while considering temperature- and salinity-related environmental data. This effectively clarified the inherent connection between genome structural variations and environmental adaptability. Additionally, this study constructed a systemic evolutionary tree using multiple marker genes and analyzed the ecological adaptability of *Synechococcus* to temperature and salinity from multiple dimensions. Moreover, it proposed a molecular mechanism for explaining how *Synechococcus* responds to environmental changes at the genomic level. At the same time, it identified several key genes and metabolic pathways that may be related to salt and temperature adaptation, providing theoretical guidance for the targeted optimization and large-scale application of *Synechococcus*.

## Figures and Tables

**Figure 1 bioengineering-10-01329-f001:**
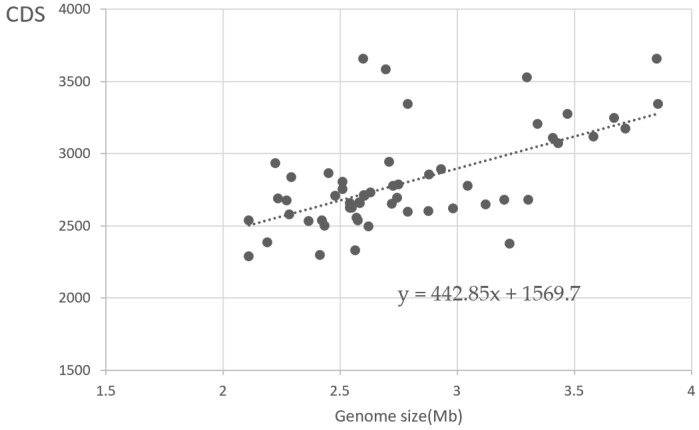
Linear regression diagram of CDS and genome size. The grey dots represent individual *Synechococcus* genomes, while the line represents the overall trend in the data.

**Figure 2 bioengineering-10-01329-f002:**
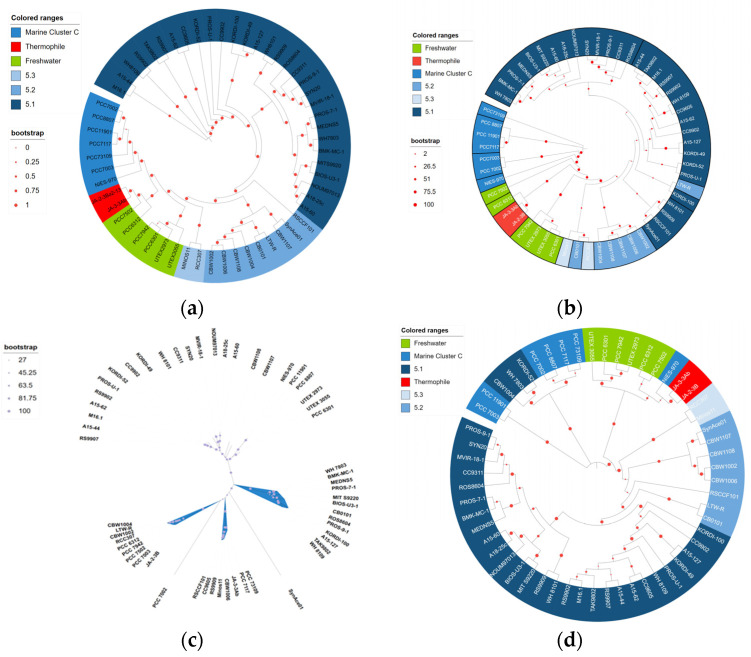
Systematic evolutionary diversity. Phylogenomic analysis with previously available marine *Synechococcus* reference genomes placed these within several clades, including 5.1, 5.2, 5.3 [17], Marine cluster C [18], thermophile, and freshwater. (**a**) Circular phylogenetic tree based on the complete genome sequence. The phylogenetic tree is from GTDB, with branch lengths reflecting phylogenetic information as inferred from the concatenation of 120 marker genes; (**b**) circular phylogenetic tree based on 16S rRNA; (**c**) unrooted circular phylogenetic tree based on peroxiredoxin, where the blue color represents the phylogenetic group to which the strain belongs; (**d**) circular phylogenetic tree analysis of cytochrome C oxidase subunit I sequences.

**Figure 3 bioengineering-10-01329-f003:**
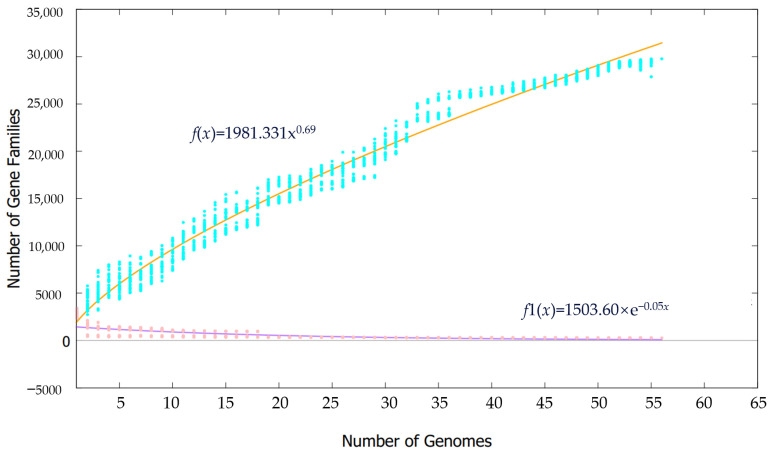
Pan and core plot of the 56 *Synechococcus* with complete genomes. The purple and orange curves represent the core genome number plot and pan-genome number plot, respectively. Equations used to fit the curves are shown above the plot, respectively.

**Figure 4 bioengineering-10-01329-f004:**
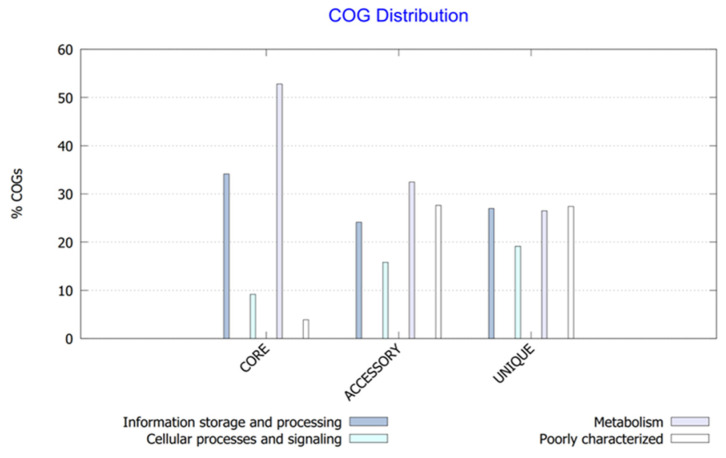
COG distribution of core, accessory, and unique genes.

**Figure 5 bioengineering-10-01329-f005:**
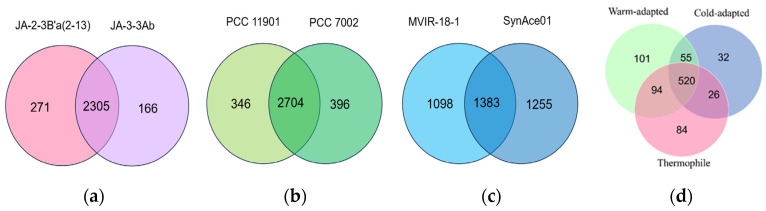
Unique KO identifiers within core identifiers. Venn diagram showing the KO identifier numbers of (**a**) thermophilic strains; (**b**) warm-adapted strains; (**c**) cold-adapted; and (**d**) three different temperature types.

**Figure 6 bioengineering-10-01329-f006:**
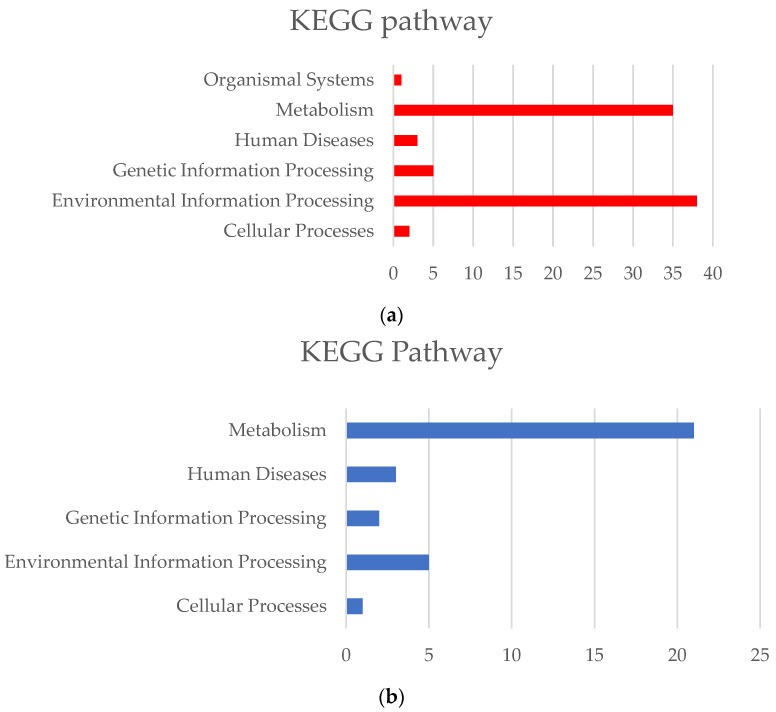
Distribution of the KEGG pathway. The red and blue bar graphs represent the quantity of KEGG pathways. (**a**) Thermophilic *Synechococcus* specific core metabolic pathway distributions; and (**b**) cold-adapted *Synechococcus* specific core metabolic pathway distributions.

**Figure 7 bioengineering-10-01329-f007:**
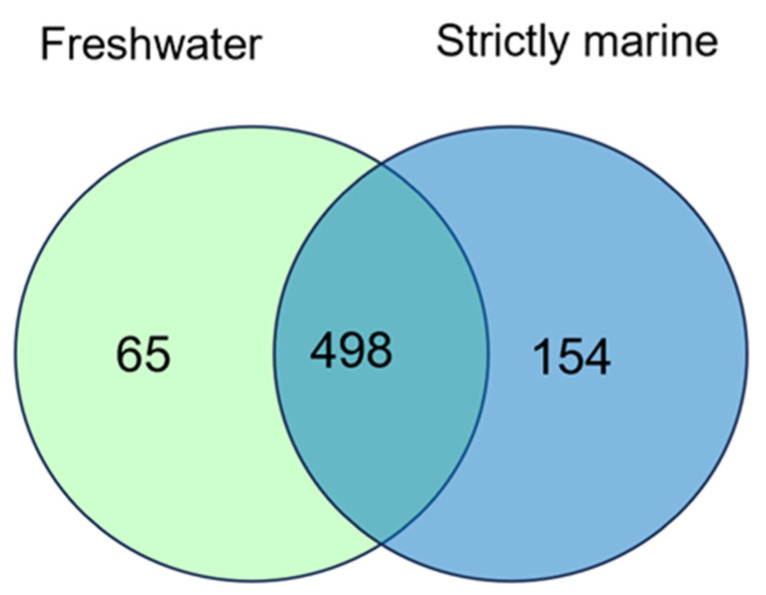
Venn diagram of pangenomes of strictly marine and freshwater strains using KEGG core KO identifiers.

**Figure 8 bioengineering-10-01329-f008:**
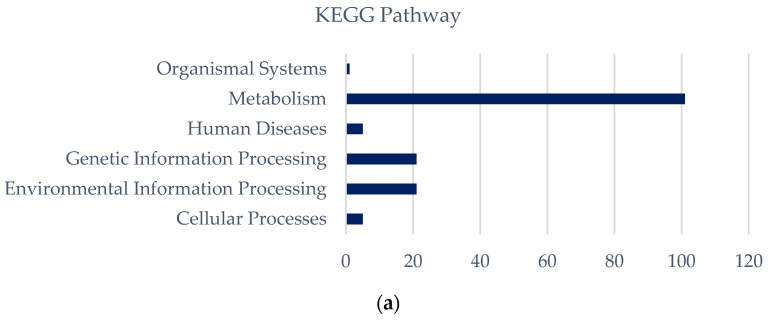
Distribution of the KEGG pathway. The deep blue and light blue bar graphs represent the quantity of KEGG pathways. (**a**) Strictly marine *Synechococcus* specific core metabolic pathway distributions; and (**b**) freshwater *Synechococcus* specific core metabolic pathway distributions.

**Table 1 bioengineering-10-01329-t001:** Genomic characteristics of 56 *Synechococcus* strains.

Organism Name	Accession	Ecosystem	Size (Mb)	GC%	CDS
*Synechococcus* sp. A15-127	NZ_CP047948.1	Marine	2.54	60.6	2653
*Synechococcus* sp. A15-44	NZ_CP047938.1	Marine	2.62	60.8	2496
*Synechococcus* sp. A15-60	NZ_CP047933.1	Marine	2.54	58.2	2623
*Synechococcus* sp. A15-62	NZ_CP047950.1	Marine	2.29	59.9	2837
*Synechococcus* sp. A18-25c	NZ_CP047957.1	Marine	2.51	58.2	2803
*Synechococcus* sp. BIOS-U3-1	NZ_CP047936.1	Marine	2.71	55	2943
*Synechococcus* sp. BMK-MC-1	NZ_CP047939.1	Marine	2.60	60.4	3654
*Synechococcus* sp. CB0101	NZ_CP039373.1	Marine	2.79	64.1	3343
*Synechococcus* sp. CBW1002	NZ_CP060398.1	Marine	3.85	64.6	3653
*Synechococcus* sp. CBW1004	NZ_CP060397.1	Marine	3.67	67.1	3247
*Synechococcus* sp. CBW1006	NZ_CP060396.1	Marine	3.86	64.6	3340
*Synechococcus* sp. CBW1107	NZ_CP064908.1	Marine	3.2	66.3	2678
*Synechococcus* sp. CBW1108	NZ_CP060395.1	Marine	3.22	63.7	2373
*Synechococcus* sp. CC9311	NC_008319.1	Marine	2.60	52.4	2707
*Synechococcus* sp. CC9605	NC_007516.1	Marine	2.51	59.2	2753
*Synechococcus* sp. CC9902	NC_007513.1	Marine	2.23	54.2	2689
*Synechococcus* sp. JA-2-3B′a(2-13)	NC_007776.1	Thermal springs	3.05	58.5	2774
*Synechococcus* sp. JA-3-3Ab	NC_007775.1	Thermal springs	2.93	60.2	2890
*Synechococcus* sp. KORDI-100	NZ_CP006269.1	Marine	2.80	57.5	2594
*Synechococcus* sp. KORDI-49	NZ_CP006270.1	Marine	2.59	61.4	2654
*Synechococcus* sp. KORDI-52	NZ_CP006271.1	Marine	2.57	59.1	2555
*Synechococcus* sp. LTW-R	NZ_CP059060.1	Marine	2.41	62.6	2297
*Synechococcus* sp. M16.1	NZ_CP047954.1	Marine	2.11	61.4	2537
*Synechococcus* sp. MEDNS5	NZ_CP047952.1	Marine	2.44	59.1	2500
*Synechococcus* sp. Minos11	NZ_CP047953.1	Marine	2.28	60.5	2577
*Synechococcus* sp. MIT S9220	NZ_CP047958.1	Marine	2.42	56.4	2537
*Synechococcus* sp. MVIR-18-1	NZ_CP047942.1	Marine	2.45	53.4	2864
*Synechococcus* sp. NIES-970	NZ_AP017959.1	Marine	3.12	49.4	2648
*Synechococcus* sp. NOUM97013	NZ_CP047941.1	Marine	2.55	58.8	2625
*Synechococcus* sp. PCC 11901	NZ_CP040360.1	Marine	3.47	49.1	3272
*Synechococcus elongatus* PCC 6301	NC_006576.1	Freshwater	2.70	55.5	3581
*Synechococcus* sp. PCC 6312	NC_019680.1	Freshwater	3.72	48.5	3171
*Synechococcus* sp. PCC 7002	NC_010475.1	Marine	3.41	49.2	3109
*Synechococcus* sp. PCC 7003	NZ_CP016474.1	Marine	3.35	49.3	3204
*Synechococcus* sp. PCC 7117	NZ_CP016477.1	Marine	3.4	49.1	3071
*Synechococcus* sp. PCC 73109	NZ_CP013998.1	Marine	3.30	49.3	3524
*Synechococcus* sp. PCC 7502	NC_019702.1	Freshwater	3.58	40.6	3118
*Synechococcus elongatus* PCC 7942	NC_007604.1	Freshwater	2.72	46.3	2649
*Synechococcus* sp. PCC 8807	NZ_CP016483.1	Marine	3.30	49.3	2679
*Synechococcus* sp. PROS-7-1	NZ_CP047945.1	Marine	2.57	59.3	2328
*Synechococcus* sp. PROS-9-1	NZ_CP047961.1	Marine	2.27	53.9	2675
*Synechococcus* sp. PROS-U-1	NZ_CP047951.1	Marine	2.57	58.4	2535
*Synechococcus* sp. RCC307	CT978603.1	Marine	2.22	60.8	2931
*Synechococcus* sp. ROS8604	NZ_CP047946.1	Marine	2.88	54	2600
*Synechococcus* sp. RS9902	NZ_CP047949.1	Marine	2.48	59.8	2706
*Synechococcus* sp. RS9907	NZ_CP047944.1	Marine	2.58	60.4	2659
*Synechococcus* sp. RS9909	NZ_CP047943.1	Marine	2.60	64.4	2711
*Synechococcus* sp. RSCCF101	NZ_CP035632.1	Marine	2.98	68	2618
*Synechococcus* sp. SYN20	NZ_CP047959.1	Marine	2.72	53.4	2773
*Synechococcus* sp. SynAce01	NZ_CP018091.1	Marine	2.75	63.9	2784
*Synechococcus* sp. TAK9802	NZ_CP047937.1	Marine	2.19	60.8	2382
*Synechococcus* sp. UTEX 2973	NZ_CP006471.1	Freshwater	2.74	55.5	2691
*Synechococcus elongatus* UTEX 3055	NZ_CP033061.1	Freshwater	2.88	55.1	2853
*Synechococcus* sp. WH 7803	CT971583.1	Marine	2.36	60.2	2533
*Synechococcus* sp. WH 8101	NZ_CP035914.1	Marine	2.63	63.3	2730
*Synechococcus* sp. WH 8109	NZ_CP006882.1	Marine	2.11	60.1	2288

**Table 2 bioengineering-10-01329-t002:** List of geographical isolation information of *Synechococcus*.

Strain Name	Geographic Origin	Isolation Latitude	Isolation Longitude	Depth (m)	Isolation Date
A15-127	North Atlantic Ocean	−31.12	−3.92	45	24 October 2004
A15-44	North Atlantic Ocean	21.68	−17.83	20	1 October 2004
A15-60	North Atlantic Ocean	17.62	−20.95	10	4 October 2004
A15-62	North Atlantic Ocean	17.62	−20.95	15	4 October 2004
A18-25c	Pacific Ocean	27.63	−37.03	74	14 September 2008
BIOS-U3-1	Pacific Ocean	−33.86	−73.34	5	28 November 2004
BMK-MC-1	Mediterranean Sea	40.8	14.15	23	13 October 2009
CB0101	Chesapeake Bay	39.28	−76.6	0	17 July 2004
CBW1002	Chesapeake Bay	39.28	−76.6	0	21 December 2010
CBW1004	Chesapeake Bay	39.28	−76.6	0	28 December 2010
CBW1006	Chesapeake Bay	39.28	−76.6	0	28 December 2010
CBW1107	Chesapeake Bay	39.28	−76.6	0	1 February 2011
CBW1108	Chesapeake Bay	39.28	−76.6	0	1 February 2011
CC9311	North Pacific, California	31.91	−124.17	95	8 April 1993
CC9605	North Pacific, California	30.42	−124	51	1 January 1993
CC9902	North Pacific, California	32.87	−117.26	5	1 January 1996
JA-2-3B′a(2-13)	Octopus Spring	22.45	−158	400	10 July 2002
JA-3-3Ab	Octopus Spring	44.55	−110.82	400	25 July 2002
KORDI-100	Pacific Ocean	9.15	158.4	0	September 2007
KORDI-49	Pacific Ocean	32.3	126	20	1 March 2004
KORDI-52	Pacific Ocean	32	126.45	30	1 May 2005
LTW-R	Pearl river, China	22.22	114.129	-	July 2014
M16.1	North Atlantic Ocean	27.7	−91.3	275	9 February 2004
MEDNS5	Mediterranean Sea	41	6	80	1 July 1993
MINOS11	Mediterranean Sea	34	18	20	2 November 2010
MIT S9220	Pacific Ocean	0	−140	0	1 October 1992
MVIR-18-1	North Atlantic Ocean	61	1.59	25	23 July 2007
NIES-970	Rikuhama Beach, Japan	-	-	-	28 August 1998
NOUM97013	Pacific Ocean	−22.33	166.33	0	19 November 1996
PCC 11901	Singapore	1.25	103.57	20	6 May 2017
PCC 6301	Austin, Texas, USA	30.41	−97.76	-	1 January 1952
PCC 6312	California, USA	-	-	-	1 January 1963
PCC 7002	Magueyes, Puerto Rico	17.96	−67.04	-	1 January 1961
PCC 7003	Greenwich, USA	-	-	-	1 January 1960
PCC 7117	Port Hedland, Australia	-	-	-	1 January 1971
PCC 73109	City Island, New York	-	-	-	1 January 1961
PCC 7502	Lake, Switzerland	-	-	-	1 January 1972
PCC 7942	California, USA	-	-	-	1 January 1973
PCC 8807	Lagoon, Gabon	-	-	-	1 January 1979
PROS-7-1	Mediterranean Sea	37.4	15.62	5	26 September 1999
PROS-9-1	Mediterranean Sea	41.9	10.43	30	28 September 1999
PROS-U-1	North Atlantic Ocean	71.22	−136.72	5	12 September 1999
RCC307	Mediterranean Sea	39.17	6.18	15	25 May 1996
ROS8604	North Atlantic	48.73	−3.98	1	24 November 1986
RS9902	Indian ocean, Red Sea	29.47	34.92	1	23 March 1999
RS9907	Indian ocean, Red Sea	29.47	34.92	10	29 August 1999
RS9909	Indian ocean, Red Sea	29.47	34.92	10	7 September 1999
RSCCF101	Indian ocean, Red Sea	29.47	34.92	20	July 2014
SYN20	North Atlantic Ocean	60.27	5.21	2	1 June 2004
SynAce01	Ace Lake, Antarctica	−65.31	78.5	15	1992
TAK9802	South Pacific Ocean	34.92	34.92	7	6 February 1998
UTEX 2973	Austin, Texas, USA	-	-	0	7 December 2011
UTEX 3055	Austin, Texas, USA	30.17	−97.44	-	13 December 2013
WH7803	North Atlantic Ocean	33.75	−67.5	25	3 July 1978
WH8101	North Atlantic Ocean	41.52	−70.67	0	1 January 1981
WH8109	North Atlantic Ocean	39.28	−70.46	-	1 June 1981

**Table 3 bioengineering-10-01329-t003:** Growth temperature characteristics of different strains of *Synechococcus*.

Strain Name	Culture Collection Number	Culture Medium	Culture Temperature (°C)	Growth Temperature (°C)	Optimum Temperature (°C)	Ecotype
A15-127	RCC 2378	PCR-S11	22	-	-	-
A15-44	RCC 2527	PCR-S11	22	25–40	30	Warm
A15-60	RCC 2554	PCR-S11	22	-	-	-
A15-62	RCC 2374	PCR-S11	22	16–32	30	Warm
A18-25c	-	PCR-S11	22	-	-	-
BIOS-U3-1	RCC 2533	PCR-S11	22	12–29	25	Cold
BMK-MC-1	-	PCR-S11	22	-	-	-
CB0101	-	SN	25	4–30	25	Cold
CBW1002		SN15	23	4–10	6.5	Cold
CBW1004	-	SN15	23	4–10	6.2	Cold
CBW1006	-	SN15	23	4–10	6.2	Cold
CBW1107	-	SN15	23	4–10	2.5	Cold
CBW1108	-	SN15	23	4–10	2.5	Cold
CC9311	RCC 1086	SN35	23	-	-	Warm
CC9605	RCC 753	PCR-S11	22	-	-	-
CC9902	RCC 2673	SN	22	10–30	24	Cold
JA-2-3B′a(2-13)	-	BG11	25	45–65	51–61	Thermophile
JA-3-3Ab	-	BG11	25	45–70	58–65	Thermophile
KORDI-100	-	BG11	28	-	29.5	-
KORDI-49	-	ASN III	20	-	13.5	-
KORDI-52	-	BG11	25	-	14.5	-
LTW-R	-	BG11	25	-	-	-
M16.1	RCC 791	PCR-S11	22	18–35	32	Warm
MEDNS5	RCC 2368	PCR-S11	22	-	-	-
MINOS11	RCC 2319	PCR-S11	22	-	-	-
MIT S9220	RCC 2571	PCR-S11	22	16–31	30	Warm
MVIR-18-1	RCC 2385	PCR-S11	15	4–25	22	Cold
NIES-970	NIES 970	ESM	20	-	-	-
NOUM97013	RCC 2433	PCR-S11	22	-	-	-
PCC 11901	PCC 11901	1536	30	20–43	38–41	Warm
PCC 6301	PCC 6301	1540	22	20–40	38	Warm
PCC 6312	PCC 6312	1539	22	20–40	38	Warm
PCC 7002	PCC 7002	1550	22	20–40	38	Warm
PCC 7003	PCC 7003	1536	22	-	-	-
PCC 7117	PCC 7117	1536	22	-	-	-
PCC 73109	PCC73109	1535	22	-	-	-
PCC 7502	PCC 7502	1539	22	-	-	-
PCC 7942	PCC 7942	1539	22	20–45	38	Warm
PCC 8807	PCC 8807	1546	22	-	-	-
PROS-7-1	RCC 2381	PCR-S11	22	-	-	-
PROS-9-1	RCC 328	PCR-S11	22	-	-	-
PROS-U-1	RCC 2369	PCR-S11	22	-	-	-
RCC307	RCC 307	PCR-S11	20	10–40	30	Cold
ROS8604	RCC 2380	PCR-S11	20	16–30	26	Warm
RS9902	RCC 2376	PCR-S11	22	-	-	-
RS9907	RCC 2382	PCR-S11	22	18–35	30–32	Warm
RS9909	RCC 2383	PCR-S11	22	-	-	-
RSCCF101	-	PCR-S11	30–38	25–38	30	Warm
SYN20	RCC 2035	PCR-S11	22	10–40	30	Cold
SynAce01	-	PCR-S11	10	−17–29.5	20	Cold
TAK9802	RCC 2528	PCR-S11	22	-	-	-
UTEX 2973	2973	BG11	20	15–41	38–41	Warm
UTEX 3055	3055	BG11	20	10–42	30	Warm
WH7803	RCC 752	PCR-S11	20	16–34	33	Warm
WH8101	RCC 2555	PCR-S11	22	16–35	25	Warm
WH8109	RCC 2033	PCR-S11	22	16–35	28	Warm

**Table 4 bioengineering-10-01329-t004:** Salinity growth conditions of the researched *Synechococcus*.

Strain Name	Size (Mb)	GC (%)	Salinity Tolerance (ppt)	Optimal Salinity (ppt)	Category
CB0101	2.78965	64.1	0–35	15	euryhaline
CBW1002	3.85412	64.6	5–22	17	euryhaline
CBW1004	3.67232	67.1	5–22	19	euryhaline
CBW1006	3.86013	64.6	5–22	19	euryhaline
CBW1107	3.20209	66.3	5–22	8	euryhaline
CBW1108	3.22622	63.7	5–22	8	euryhaline
CC9311	2.60675	52.4	24–44	33.3	strictly marine
CC9605	2.51066	59.2	24–44	34	strictly marine
CC9902	2.23483	54.2	30–44	35	strictly marine
KORDI-100	2.789	57.5	-	34.2	strictly marine
KORDI-49	2.58581	61.4	-	34.2	strictly marine
KORDI-52	2.57207	59.1	-	33.8	strictly marine
LTW-R	2.41553	62.6	14–44	24–34	euryhaline
PCC 11901	3.47181	49.1	0–100	25	euryhaline
PCC 6301	2.69625	55.5	-	-	freshwater
PCC 6312	3.7205	48.5	-	-	freshwater
PCC 7002	3.40993	49.2	0–90	0–25	euryhaline
PCC 7003	3.34509	49.3	-	-	strictly marine
PCC 7117	3.4321	49.1	-	-	euryhaline
PCC 73109	3.29868	49.3	-	-	euryhaline
PCC 7502	3.58373	40.6	-	-	freshwater
PCC 7942	2.72155	46.3	0–29	23	freshwater
RCC307	2.22491	60.8	-	37	strictly marine
SynAce01	2.75063	63.9	10–50	20–30	euryhaline
UTEX 2973	2.74463	55.5	6–12	9	freshwater
UTEX 3055	2.88122	55.1	-	-	freshwater
WH7803	2.36698	60.2	24–44	34	strictly marine

## Data Availability

Data are available from the corresponding authors upon reasonable request.

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
