# Peer review of "Genomic Insights on the Carbon-Negative Workhorse: Systematical Comparative Genomic Analysis on 56 Synechococcus Strains"

_bioengineering, 2023, doi:10.3390/bioengineering10111329_

Round 1

Reviewer 1 Report

Comments and Suggestions for Authors

It is difficult for me to assess the value of this study to the scientific community but it is soundly conducted.

Major comments

L114. Compared to other cyanobacteria. Given how many there are I don’t think you can say this. I would remove the whole sentence.

Table 2. UTEX 2973 was not isolated from the Sargasso sea but from a creek near Austin, Texas. This is the strain it was streaked from: https://utex.org/products/utex-0625?variant=30991216967770

L242. GlgC, glgA and GlgB are not involved in glycogen degradation and biosynthesis. The GlgX and GlgP proteins degrade glycogen. You partially say this in the next sentence. Correct this. https://portlandpress.com/bioscirep/article/40/4/BSR20193325/222317/Current-knowledge-and-recent-advances-in

L344. The carbonic bicarbonate transport system is not regulated by CmpA, B and C but composed of these subunits, in addition to CmpD. (see ref above). Correct this.

Minor comments. Please make these changes to the manuscript.

L39. Start sentence with ‘Many Synechococcus species have key advantages…’

L107. Change to ‘They have a wide …’

L111. GC content

L383. Remove substrate

Comments on the Quality of English Language

See above

Reviewer 2 Report

Comments and Suggestions for Authors

In this manuscript, Qian et al. have analyzed the genomes of 56 Synechococcus strains available in public databases to explain the adaptability of the strains to their niche. While the study may be of interest if conducted or presented in some more detail, as presented it appears shallow with little useful information. Overall, it is not clear if the genes discussed are specific to the genomes of the sub-group whose interesting property is being investigated (e.g., higher temperature tolerance). As such the genes discussed are too generic and well-known.

Major comments

1.       The abstract is poorly written and provides little insight into what was specifically done and found. The authors mention genome-scale metabolic models in the abstract. However, no specific results or methods in relation of either reconstruction or analysis of GSMMs are presented.

2.       The English language needs to be improved throughout the manuscript.

3.       In Fig 2, what are the clusters 5.1 to 5.3?  

4.       On line 168, it is written that these strains were further classified into cold-adapted, warm-adapted, and thermophilic strains based on their minimum tolerance for growth temperature. However, Table 3 shows other Ecotypes such as coastal and tropical etc. It is not clear how many categories were actually used in the analysis.

5.       Section 3.3.1, line 182: “These strains also contain acetyl-CoA carboxylase (ACC), which catalyzes fatty acid synthesis”. This is very generic, and incorrect statement. ACC catalyze the first step in fatty acid synthesis. Also, do only these strains contain ACC? If not, is there any specific mutation in ACC in these strains? Otherwise, what is the point of discussing ACC which is a well-known initiator of fatty acid synthesis.

6.       Similarly, in section 3.3.2, it is not clear if the stated proteins are exclusive to the heat-adapted cells.

7.       Section 3.3.4: On line 230, it is written “… cold-adapted spirulina strains…” I believe that the authors mean Synechococcus strains. Here too, it is not clear that the stated proteins (LcyB and glg proteins, etc.) were exclusive to cold-adapted Synechococcus. If they are not exclusive, why were they discussed here?

8.       Section 3.4.2: Same comment as above: are the discussed genes exclusive to the cold-adapted genomes?

Comments on the Quality of English Language

The English language needs to be improved throughout the manuscript. There are several instances of unnecessarily long and complex sentences. At many other places, the sentences are not clear.

Reviewer 3 Report

Comments and Suggestions for Authors

This manuscript reports a phylogenetic and comparative genome assessment on 56 strains of Synechococcus. Key metabolic pathways and genes are reported for 6 representative strains and has   a focus on temperature and salt/fresh-water adaptation mechanisms.  The study is comprehensive and well-structured providing useful information on taxonomic relationships and metabolic function for Synechococcus.

The title of the paper does not seem appropriate. I think the term “carbon negative workhorse” should be deleted. Unless the paper is re-written to show that somehow Synechococcus is carbon negative – that would be very challenging.  A better title could be just the second half i.e. “A comparative Genomic Analysis on 56 Synechococcus strains”.

The Abstract needs significant improvement.

Currently at least half the first part of the abstract covers background and some of this does not seem relevant or justified. The paper is not focussed on synthetic biology so better not to include this in the first sentence. Also, carbon fixation and nitrogen fixation are not relevant to this study. Likewise, the paper does not cover how these organisms are carbon negative.  

Some of the terminology used does not seem appropriate and is difficult to understand – for example what is meant by “fully matured operational facilities” or "indispensable". 

Most of the first half of the Abstract can be deleted. The second half of the abstract can then be expanded to focus on the reported study and key findings.  

The Introduction is satisfactory but could usefully provide more information on the chronological order on collection/identification of species.  

Again, there are several instances of misleading terminology and not being literally correct or are too vague.  For example:  

1. Synechococcus…. “capable of easily addressing various environmental challenges.”

2. “Can be found almost everywhere imaginable”,

 3. “Meanwhile, synthetic biology provides powerful technical support for the industrialization of Synechococcus.”

There is introduction about synthetic biology. If this is to remain then it should be followed up with discussion on how the results can be used in synthetic biology.  Currently including this in the Introduction is not relevant. 

It would be helpful to have full species names, where available, alongside strain numbers. e.g Synechococcus elongatus, Synechococcus nidulans etc. e.g. both PCC7002 and WH8102 strains are Synechococcus elongatus   

Reference for inclusion:  Komárek et al 2020. Phylogeny and taxonomy of Synechococcus–like cyanobacteria. DOI: 10.5507/fot.2020.006.

In the Methods section, at least, it would be useful to include more detailed information on where only 6 strains were used for analysis and why these six strains were selected.

The Results section and associated Tables and Figures including those in the Supplementary, represent the main strength of the manuscript; they are well structured and provide useful information.  

The Discussion has scope for expansion with further discussion on some findings from the results. It would be useful to have some examples of how the results could be used in metabolic engineering.  If mention of synthetic biology is to be kept in the introduction (I recommend deleting this) then the authors should discuss the potential of their results in terms of synthetic biology. 

Typo spotted: Line 266 that of euryhaline strains.  

Comments on the Quality of English Language

I think some of the terminology used in the Abstract and Introduction relates to the quality of the English Language leading to meaning of some phrases not being scientifically sound.    

Round 2

Reviewer 2 Report

Comments and Suggestions for Authors

The authors have satisfactorily addressed my comments. The article can be accepted for publication.